# Nitrogen Source Preference and Growth Carbon Costs of *Leucaena leucocephala* (Lam.) de Wit Saplings in South African Grassland Soils

**DOI:** 10.3390/plants10112242

**Published:** 2021-10-21

**Authors:** Nonkululeko Sithole, Zivanai Tsvuura, Kevin Kirkman, Anathi Magadlela

**Affiliations:** 1School of Life Sciences, College of Agriculture, Engineering and Science, University of KwaZulu-Natal, Westville Campus, Private Bag X54001, Durban 4000, South Africa; lulekosithole@gmail.com; 2School of Life Sciences, College of Agriculture, Engineering and Science, University of KwaZulu-Natal, Pietermaritzburg Campus, Private Bag X01, Scottsville 3209, South Africa; Tsvuuraz@ukzn.ac.za (Z.T.); KirkmanK@ukzn.ac.za (K.K.)

**Keywords:** emerging invasive, KZN soils, *Leucaena leucocephala*, N fixation, N and P deficiencies

## Abstract

*Leucaena leucocephala* (Fabaceae) is native to Central America and has invaded many climatic regions of the tropics. In South Africa, the species is categorized as an emerging or incipient weed used as fodder, timber, firewood and in erosion control on degraded habitats. The species is common along the eastern subtropical regions of KwaZulu-Natal (KZN) Province, where it invades grasslands, savannas and edges of forests. Soils of these ecosystems are characterized as nutrient deficient and acidic. Using a pot trial, we determined the effects of the nutrient addition treatments on microbial symbiosis, N nutrition and biomass accumulation of *L. leucocephala* under greenhouse conditions. After 180 days of growth, plants were harvested, and their utilization of N derived from the atmosphere and from the soil was quantified through determination of *δ^15^*N values. *L. leucocephala* maintained growth and N nutrition by relying on both atmospheric- and soil-derived N across all soil treatments. The NDFA was significantly higher in high P (N1 + P, N2 + P and N3 + P) soils. *L. leucocephala* was able to nodulate with intermediate and fast-growing strains from the *Mesorhizobium* and *Rhizobium* genus in N2 + P grown plants. This shows that *L. leucocephala* possesses traits that are successful in acquiring nutrients, especially in nutrient limited conditions, by establishing plant symbiosis with multiple bacteria and relying on extracting N from the soil and from the atmosphere through the symbiosis.

## 1. Introduction

*L. leucocephala* (Lam.) de Wit (Fabaceae) is native to Central America and has been introduced in many geographic and climatic regions [1]. The species has been listed in the top 100 worst invaders of the world following deliberate introductions for agroforestry [1]. In South Africa, *L. leucocephala* is categorized as an emerging or incipient weed mostly occurring in the eastern subtropical parts of the country [2] made up of grassland, forest and savanna. However, most invasions have been noted in the savanna biome [3]. Savanna and grassland ecosystems provide multiple ecosystem services [4] and contribute to the rural and agricultural economy of South Africa in various ways [5], and the rural socio-economy [6]. Therefore, sustainable management of plant biodiversity is an integral part of these ecosystems. South African savanna and grassland ecosystems are reported to be nutrient limited, especially with regards to nitrogen (N) and phosphorus (P) [7,8]. In addition, the soils are also acidic [9]. Legume plants require N and P for various processes such as nodule growth and function [10]. Despite the nutrient limitations in the savanna and grassland ecosystems, the legume taxa continue to thrive and are most represented among the invasive species in nutrient poor ecosystems [11]. N-fixing bacteria and arbuscular mycorrhizal (AM) fungi symbiosis has been recognized as the driver of legume invasions as they can acquire N and P, respectively [12,13]. The transfer of fixed carbon (C) from the host to the symbiont has a direct effect on the host plant, and, thus, it is important to quantify this process [14,15,16]. Generally, it is more cost-effective for plants to assimilate soil inorganic N than atmospheric N_2_ due to the C costs to symbionts [17]. Legumes can switch their N preferences due to nutritional stress, favoring soil N uptake to reserve energy [16,17,18]. Therefore, the availability and assimilation of soil inorganic N can reduce the cost of N_2_ fixation. In addition to symbiotic N fixation, non-symbiotic N fixation also plays an essential role in the grassland N economy [19].

Invasive legumes can efficiently exploit scarce nutrients and yield an aboveground biomass rich in N better than their neighboring native species. Consequently, N increases beyond levels at which the indigenous species are adapted to thrive [20,21], and so they may be displaced by invasive species due to their aggressive growth [22]. The N contributed by the biological nitrogen fixation (BNF) process from indigenous legumes is less than that produced by invasive legumes due to their slow growth rates and the absence of competition from natural enemies [23]. For example, the invasive legume from tropical Africa, *Senna didymobotrya*, can efficiently acquire more nutrients than native plants in South African poor soils [24]. This leads to the question of whether these invasive plants have traits or mechanisms that enhance their competitive ability for nutrient uptake and conversation [25]. It has been stated that invasive legumes nodulate readily using both native and non-native rhizobia species and are considered prolific N_2_-fixing species [26]. Rodríguez-Echeverría et al. [13] reported that *Acacia longifolia* was more efficient at forming a symbiotic association with bacteria and fixed more N than other co-occurring N_2_ fixing legumes, which was similar to that reported in the Cape fynbos [21]. Consequently, for the first time, this study investigates the plant–microbe symbiosis, plant nutrition, C costs and biomass accumulation in *L. leucocephala* grown in acidic grassland soils with varying N and P nutrient status. The proposed hypothesis was that *L. leucocephala* would establish plant–microbe symbiosis with multiple and more efficient N-fixing bacteria and change its N source preference to reduce growth C costs in P deficient soils.

## 2. Results

### 2.1. Soil Characteristics

The average percentage N concentration was 11% lower in N1 and N2 + P soils compared to the average percentage N concentration of N2, N3, N1 + P and N3 + P soils (Appendix A). The average percentage P concentration was 57% higher in high P and N2 soils compared to the average percentage P concentration of N1 and N3 soils. Conversely, the average percentage K concentration was 51% higher in N1 and N3 soils compared to the average percentage K concentration of other experimental soils (high P and N2) (Appendix A). The average percentage exchangeable acidity was 74% higher in N3 and N3 + P soils compared to the average percentage exchangeable acidity of N1, N1 + P and N2 + P soils. All of the soils were acidic with a pH (KCl) below 5 except N2 + P, which had the pH (KCl) of 5.01. The pH (KCl) and pH (H_2_O) in N3 and N3 + P soils were 11% and 16% more acidic than other soils, respectively (Appendix A).

### 2.2. Soil Bacteria and Plant Endophytic Bacterial Isolates Identification

The molecular identification of N-fixing and N-cycling bacteria in the experimental soils used as growth substrate included *Caulobacter rhizosphaerae*, *Sphingomonas* sp. and *Burkholderia contaminans*, with accession no. and similarity (%) detailed in Appendix A. Only the *L. leucocephala* plants grown in N2 + P soils were able to form a symbiotic association with the N-fixing bacteria. The 16S rRNA gene revealed that the symbionts were various intermediate and fast-growing strains from the *Mesorhizobium* and *Rhizobium* sp. in plants grown in this treatment (Appendix A) even though the *Mesorhizobium and Rhizobium* species were present in all soils.

### 2.3. Biomass and Mineral Contents

*L. leucocephala* plants grown in N + P soils had significantly higher total biomass than plants grown in other soils (Table 1). Leaf biomass was significantly higher in N1 and N + P soils (Table 1). Shoot and root biomass showed a similar trend of significantly higher values in the +P treatments than in the N only treatments (Table 1). The root–shoot ratio of the plants grown in N3 + P soils was significantly higher compared to the N2 and N3 soils (Table 1). The P concentration was significantly higher in *L. leucocephala* grown in the N1 and in the +P soils than in the N2 and N3 soils. Inversely, the plant N concentration was significantly higher in the N2 and N3 soils than in the N1 and N + P soils (Table 1).

### 2.4. Growth Rates and Carbon Construction Costs

*L. leucocephala* plants grown in the N1 and N + P soils attained significantly higher growth rates compared to *L. leucocephala* grown in N3 soils (Figure 1A). A similar trend was observed in the relative growth rate, which was also higher in these treatments (Figure 1B). In relation to the carbon construction costs, the N3 and N2 + P soils presented the highest values when compared to the N1 + P and N3 + P soils (Figure 1C).

### 2.5. N and P Use Efficiencies

Plants grown in N2 and N3 soils showed increased specific N assimilation rate (SNAR), while plants grown in other soils had lower values (Figure 2A). The specific N utilization rate (SNUR) was much greater in the N + P soils than in the other soils (Figure 2C). The specific P assimilation rate (SPAR) was significantly greater in *L. leucocephala* plants grown in the N1 and N+ P soils than in plants of other soils (Figure 2B), while the specific P utilization rate (SPUR) was significantly higher in *L. leucocephala* plants grown in N + P soils than plants in other soils (Figure 2D).

### 2.6. N Preference

Plants grown in the N + P soils relied more on NDFA, whereas the low P soil grown plants (i.e., the N only treatments) relied more on NDFS (Figure 3). 

## 3. Discussion

*Leucaena leucocephala* was able to grow in acidic KZN grassland soils with variable N and P by regulating their growth, nutrient assimilation and utilization rates. Moreover, the *L. leucocephala* plants grown in soils with moderate N and high P (N2 + P) concentrations were the only plants that established a symbiosis with the N-fixing bacteria. The 16S r RNA gene revealed that several species from the *Mesorhizobium* and *Rhizobium* genus inhabited the nodules of *L. leucocephala* grown in N2 + P soils. Non-nodulations in the *L. leucocephala* plants grown in the P deficient, low N and high N with P soils could be attributed to the low pH levels (pH (KCl) 4.12–4.67) observed in these soils compared to pH (KCl) 5.01 in N2 + P soils. Extreme low pH levels have been reported to reduce approximately 90% nodulation in various species such as cowpea, pea, *Lucerne* and soybean [27], affecting both indeterminate and determinate nodules. For example, acidity disrupts the signaling mechanisms between the *Rhizobium* and the plant roots, as explained in detail by [27,28].

In addition, nodule formation and plant growth in *L. leucocephala* plants grown in N3 soils could have been negatively affected by increased ammonium (NH_4_^+^) in the soil [29]. NH_4_^+^ is a paradoxical nutrient: it is beneficial due to its readily oxidative state for plant assimilation but can cause toxicity symptoms when available in large amounts. Therefore, increased soil N fertilization, and soil acidity may have reduced the total biomass of N3 grown *L. leucocephala*. This is highlighted by the increase in specific N assimilation rates, decreased specific N utilization rates and plant growth rates in N3 soil grown plants. Limited soil P may have also contributed to the decreased biomass as P is the second most essential nutrient required for plant growth and development [30]. In P-limited and acidic soil conditions, some legumes often prefer NDFS to NDFA as it is assumed that biological N fixation (BNF) requires significantly high amounts of energy (ATP) to fix one molecule of N compared to the energy required for the uptake and reduction in NO_3_ [31,32]. This could explain the observed N derived from soil results across all soils. The percentage NDFA was high in nodulated and non-nodulated *L. leucocephala* plants grown in P fertilized soil, showing the significance of P as an energy driver in this process [31,32]. In addition to symbiotic N fixation, invasive legumes also depend on non-symbiotic N fixation by bacteria species [33] as they supply N in both organic and inorganic form [34]. Therefore, non-nodulation and NDFA in the *L. leucocephala* plants suggest an association with non-nodulating endophytic or associative rhizosphere N-fixing bacteria species. The presence of bacteria from the *Burkholderia* and *Caulobacter* genus in the soil has been reported to improve plant and soil health by supplying urea-N from BNF [35]. *Sphingomonas* sp. has been isolated in barley, millet and wheat and reported to fix atmospheric N, and has been classified as plant growth-promoting bacteria [32]. N-fixing and N-cycling bacteria (*Caulobacter rhizosphaerae*, *Burkholderia contaminans* and *Sphingomonas sp.*) were identified in the experimental soils used as growth substrate. Even with the increased NDFA in high P soils, *L. leucocephala* plants utilized N derived from the atmosphere and N from the soil (NDFS).

N and P often limited plant growth in grassland ecosystems as plants often increase growth when both N and P are added in soils [36]. This was also observed in the current study as the *L. leucocephala* plants grown in N + P soils had higher total biomass than the *L. leucocephala* plants grown in -P soils. Surprisingly, *L. leucocephala* grown in N1 soils accumulated more total biomass and increased growth rate compared to plants grown in N2 and N3 soils regardless of the soils having a significantly low concentration of N and P. This could be attributed to the various adaptations displayed by plants during P deficiency. These adaptations include investing more resources on below ground biomass to increase the root surface area for nutrient absorption [37]. This might have been the case in N1 grown plants as the root biomass was significantly greater. Changes in root architecture due to nutrient deficiency are reported to increase P acquisition through increased mining of limiting nutrients in the rhizosphere [37]. This concurs with our findings as N1 grown *L. leucocephala* had an increased specific phosphorus absorption rate (SPAR) and specific phosphorus utilization rate (SPUR) coupled with an increased P content. The study conducted was important in providing a general understanding of *L. leucocephala* response to nutrient variability associated with acidity in grassland ecosystem soils. Further research on the interactions of *L. leucocephala* with native legume plants in these ecosystems is pertinent to generate more information to link to the physiological adaptations of *L. leucocephala*.

## 4. Materials and Methods

### 4.1. Study Site

Soil samples were collected from the Veld Fertilizer Trial (VFT) at Ukulinga (29°24′ E, 30°24′ S; altitude 847 m above sea level), a research farm of the University of KZN in Pietermaritzburg, South Africa. The mean precipitation and temperature of the area is approximately 838 mm and 18 °C, respectively [38]. The vegetation at Ukulinga is described as KwaZulu-Natal Hinterland Thornveld [39], which is an open savanna dominated by tall C4 grasses such as *Themeda triandra*, *Hyparrhenia hirta* and *Heteropogon contortus* while the sparse tree layer is dominated by *Vachellia sieberiana* and *V. nilotica*. Soils are deep (600–1000 mm) dolerites and shales derived from Karoo sediments of the Westleigh form [39].

### 4.2. Experimental Design

The VFT was initiated in 1951 through the addition of fertilizer (nitrogen (N), phosphorus (P)) and lime (L) to improve grassland productivity. There were initially 96 plots from 1951–2019 and each plot was 9.0 m × 2.7 m^2^ in size with a 1 m spacing between plots. The VFT experiment was replicated in three blocks, each block containing 32 plots, resulting in a 4 × 23 treatment structure laid out in a complete randomized design. For the purposes of this study, we used treatment plots fertilized with N in the form of limestone ammonium nitrate (LAN) and P in the form of superphosphate. Three levels of 28% LAN (N1 = 210 kg/ha/season; N2 = 421 kg/ha/season and N3 = 632 kg/ha/season) fertilizer. In addition, the three N levels were also applied in combination with one level of 11.3% superphosphate (336 kg/ha/season) (N1 + P, N2 + P and N3 + P). This completely randomized block design experiment for this study adds up to six treatments.

### 4.3. Soil Characteristics Analysis and Bacterial Identification

For each of the six treatments, five soil samples were obtained from each plot in the three blocks to a depth of 30 cm. Thereafter, the soils for each treatment were pooled for uniformity. Five subsamples of 50 g soil from each treatment were sent for analysis, which included nutrients such as P, N, K and other soil properties such as pH, acidity exchange and total cation at the KZN Department of Agriculture and Rural Development Analytical Services Unit at Cedara, South Africa. In addition to the soil characteristics, soil moisture factor was also accounted for by drying five soil samples from each treatment in an oven at 105 °C until a constant weight was achieved, as detailed by [40]. An additional 5 soil samples (250–300 g) from each treatment were used for microbial identification, where the bacterial template DNA was extracted using a modified boiling procedure, by boiling 300 µL of bacterial culture in 10% TSA suspended in Milli-Q water in a safe-lock Eppendorf tube for 10 min, cooled on ice and centrifuged as described by [41]. The bacterial DNA amplification using the 16S rRNA gene, sequencing and identification was performed as detailed in [42].

### 4.4. Seed Collection, Germination and Growth Conditions

Seeds of *L. leucocephala* were collected from randomly located trees at Roosfontein Nature Reserve, Durban, South Africa. The experiment was conducted under ambient conditions in a greenhouse at the University of KZN botanical garden at Pietermaritzburg, South Africa. The conditions in the greenhouse were: day-time temperatures of 12 to 14 °C and night-time temperatures of 30 to 35 °C with humidity from 70% to 80% and irradiance ~35% of full sunlight (i.e., 415.6 µmol m^−2^ s^−1^). Prior to germination, the seeds were soaked in 15% sodium hypochlorite for 20 min. Thereafter, they were rinsed five times in distilled water and then placed in petri dishes layered with Whatman’s filter paper for germination. The seeds were watered every day until seedling emergence (10 days). Thereafter, in 15 cm diameter pots, seedlings were planted at a depth of ~2 cm. Each soil treatment had 20 replicates. Plants were irrigated every 2 days in the afternoon depending on the climatic conditions.

### 4.5. Plant Harvesting and Nutrient Analysis

The initial harvest of five plants from each treatment for the initial values required in the growth calculations took place after 30 days and final harvests of 10 plants from each treatment took place 180 days after seedling emergence. At each harvest time, plants were rinsed with distilled water then separated into leaves, stems, roots and nodules, and, thereafter, oven dried at 65 °C for 4 days before weighing and grinding to a powder. The ground plant material was stored in 2 mL Eppendorf tubes and was sent for C and isotope N analysis at the Archaeometry Department, University of Cape Town, and for P analysis at the Central Analytical Facilities at Stellenbosch University, both in South Africa. Five remaining plants from the N2 + P treatment were nodulated, root nodules were harvested for bacterial extraction. Root nodules were rinsed with distilled water, then sterilized in ethanol 70% (*v*/*v*) for 30 s and with 3.5% (*v*/*v*) sodium hypochlorite solution for 3 min, and, thereafter, rinsed 10X with distilled water then stored in airtight vials containing silica gel and cotton wool. The vials were then stored at 4 °C for bacterial extraction, culturing in yeast mannitol agar (YMA) and sequencing.

### 4.6. Bacterial Extraction and Identification

Prior to bacterial extraction, the nodules were transferred into 2 mL Eppendorf tubes containing distilled water and left overnight to absorb water at 4 °C. The nodules were again sterilized in ethanol 70% (*v*/*v*) for 30 s and with 3.5% (*v*/*v*) sodium hypochlorite solution for 3 min. Thereafter, nodules were rinsed 10X with distilled water. The second sterilization was to remove any contaminants that might have been introduced during storage. The nodule samples were then crushed in 15% glycerol solution. The turbid nodule solution in 15% glycerol was streaked in plates containing yeast mannitol agar (YMA) containing 0.5 g/L yeast extract (Glentham Life Sciences Ltd., Corsham, UK), 10 g/L mannitol (Merck KGaA, Darmstadt, Germany), 0.5 g/L di-potassium hydrogen orthophosphate (K_2_HPO_4_, Merck KGaA, Darmstadt, Germany), 0.2 g/L magnesium sulfate heptahydrate (MgSO_4_.7H_2_O, Merck KGaA, Darmstadt, Germany), 0.1 g/L sodium chloride (NaCl, Merck KGaA, Darmstadt, Germany), 15 g/L bacteriological agar (Merck KGaA, Darmstadt, Germany) and incubated at 28 °C. The bacteria were re-streaked into fresh plates until pure colonies/cultures were obtained.

The pure bacterial colonies/cultures randomly selected based on phenotypes were amplified using a portion of 16-S rRNA gene, 27F (5′-AGAGTTTGATCCTGGCTCAG-3′) and 1492R (5′-GGTTACCTTGTTACGACTT-3′). Bacterial DNA amplification was performed following the protocol: initial denaturation for 5 min at 94 °C, 30 cycles of denaturation 30 s at 94 °C, annealing at 55 °C and elongation for 2 min at 72 °C, followed by a final elongation step of 10 min at 72 °C using a Bio-Rad Mini Opticon Thermal cycler (Bio-Rad Laboratories Ltd., Rosebank, Gauteng, South Africa). In total, 25 µL PCR reactions contained 11 µL sterile distilled water, 12.5 μL TAKARA-EmeraldAmp GT PCR Master Mix (Separations, Randburg, Gauteng, South Africa), 0.25µL 27F primer, 0.25 µL 1492R primer and 1 µL of DNA colony. The results were viewed in 1% (*m*/*v*) agarose gel electrophoresis using TAE buffer and run at 100 V for 20 min. Thereafter, amplified products were sent for sequencing at the Central Analytical Facilities. The resulting sequences were edited and subjected BLASTN (National Center for Biotechnology Information, NCBI, https://www.ncbi.nlm.nih.gov/genbank/ accessed on 16 June 2020) to compare them to all of the other bacterial 16S rRNA sequences already in the database.

### 4.7. Growth Calculations

#### 4.7.1. Calculation of the Specific N/P Absorption and Utilization Rates

Specific N and P absorption rate (SNAR) values were calculated according to [43] by calculating the total N and P absorbed/assimilated by the plant roots (mg N/P g^−1^root DW day^−1^)
SNAR = (N_2_ − N_1_/t_2_ − t_1_) ∗ [(log_e_ R_2_ − log_e_ R_1_)/(R_2_ − R_1_)](1)
SPAR = (P_2_ − P_1_/t_2_ − t_1_) ∗ [(log_e_ R_2_ − log_e_ R_1_)/(R_2_ − R_1_)](2)
where N and P denote the total nitrogen and phosphorus content in the plant, respectively, t_2_ − t_1_ is the difference in time between the final and initial harvest and R_2_ and R_1_ are the final and initial root dry weight, respectively, as described in [43].

Specific N and P utilization rate (SNUR) values were calculated according to [43] by calculating the dry weight acquired by the plant during nitrogen uptake (g DW · mg^−1^ N/P · day^−1^)
SNUR = (W_2_ − W_1_/t_2_ − t_1_) ∗ [(log_e_ N_2_ − log_e_ N_1_)/(N_2_ − N_1_)](3)
SPUR = (W_2_ − W_1_/t_2_ − t_1_) ∗ [(log_e_ P_2_ − log_e_ P_1_)/(P_2_ − P_1_)](4)
where W is the plant’s dry weight [43] and the other parameters are as defined in the SNAR equation.

Relative growth rate was calculated according to [44] by calculating the total plant dry weight increase in a time interval in relation to the initial weight or dry matter (ln DW increase · day^−1^).
RGR = [(ln W_2_ − ln W_1_)/(t_2_ − t_1_)](5)
where W denotes the dry weights (DW) of the final and initial harvest and t_2_ − t_1_ is the difference in time between the harvests.

#### 4.7.2. Carbon Construction Costs

Carbon construction costs (C_w_) (mmol C g^−1^ dry weight (DW)) were calculated from Mortimer et al. [45] as derived from Peng et al. [46] as follows
C_w_ = (C + kN/14 ∗ 180/24) (1/0.89) (6000/180)(6)

C_w_ denotes tissue total carbon construction cost, C is the total concentration of carbon (mmol C g^−1^), k is the reduction state of N substrate (for NH_3_ = −3) and N is the total organic nitrogen content of the tissue (g DW^−1^), as described by [47]. The value 14 is the atomic mass of N, 180 is a conversion factor from moles to grams of glucose, the amount of electrons in a glucose molecule that are available is 24, 0.89 is an estimate of growth efficiency [47] and the fraction 6000/180 is a constant conversion factor from g^−1^ dry weight to mmol C g^−1^ DW for glucose.

#### 4.7.3. Determination and Calculation of N Derived from the Atmosphere

Isotopic N was analysed following the protocols at the Archaeometry Department, University of Cape Town where samples were combusted in a Fisons NA 1500 (Series 2) CHN analyser (Fisons Instruments SpA, Milan, Italy). Isotope N values for the N gas released were determined using the Finnigan Matt 252 mass spectrometer (Finnigan MAT GmbH, Bremen, Germany), connected to a CHN analyser by a Finnigan MAT Conflo control unit. A total of five standards were used to correct the samples for machine drift. The N isotopic ratio was calculated as *δ* = 1000 (*R*_sample_/*R*_standard_), where *R* is the molar ratio of the heavier to the lighter isotope of the samples and standards.
%NDFA = 100 ((*δ^15^*N reference plant − *δ^15^*N legume)/(*δ^15^* N reference plant − *β*))(7)
where NDFA is the N derived from the atmosphere. The *β* value represents the *δ^15^*N natural abundance of the N derived from biological N_2_ fixation, which for *L. leucocephala* plants grown in N free culture was determined to be −2.60‰.

## 5. Statistical Analysis

The effects of N and P concentration variability in the VFT on plant biomass, plant mineral nutrition and growth kinetics were examined by one-way analysis of variance (ANOVA), followed by Tukey’s HSD post hoc tests. (*p* < 0.05)., Where the assumptions of normality were not satisfied, a Kruskal–Wallis test was performed. Statistical analysis was performed using SPSS Statistics for windows v. 26 (IBM Corp., Armonk, NY, USA). 

## 6. Conclusions

*Leucaena leucocephala* plants grown in these VFT soils utilized atmospheric- and soil-derived N across all treatments. This invasive legume plant established symbiosis in less acidic soils with intermediate and fast-growing strains from the *Mesorhizobium* and *Rhizobium* spp in moderate N and high P soils. The non-nodulation and reliance on atmospheric N of the invasive legume grown in other soil treatments are associated with rhizospheric free-living N-fixing and N-cycling bacteria identified in the experimental soils. This shows that *L. leucocephala* possesses traits that allow it to successfully acquire nutrients, especially in nutrient limited conditions, by investing C resources on below ground biomass, altering N sources and nutrient assimilation and utilization rates, and establishing symbiosis with nodule forming and non-nodulating endophytic or associative rhizosphere N-fixing bacteria species.

## Figures and Tables

**Figure 1 plants-10-02242-f001:**
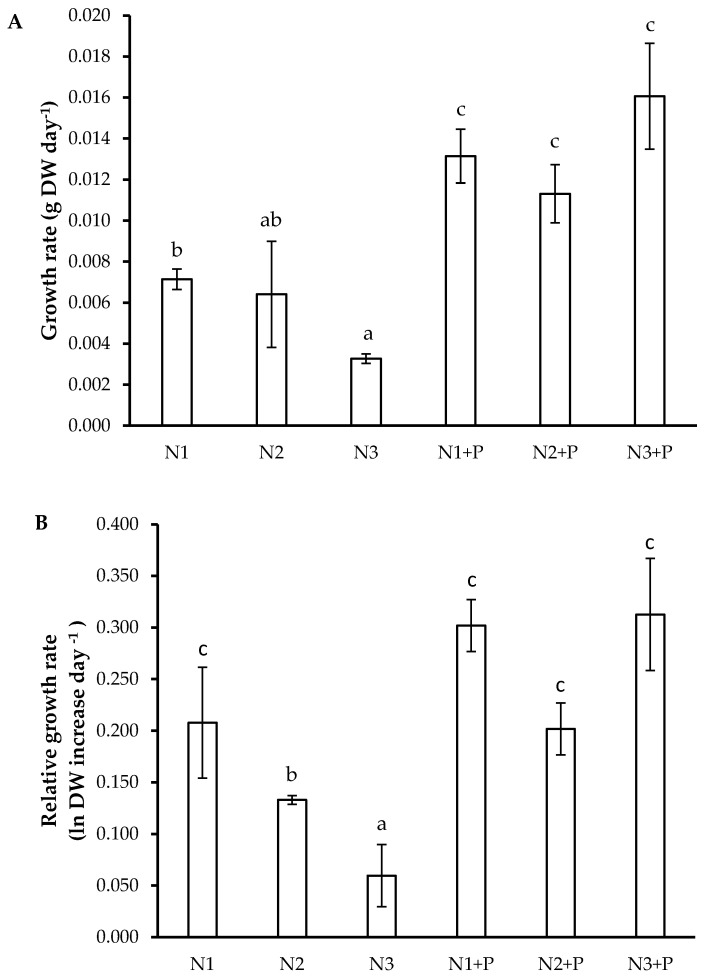
(**A**) Growth rate (g DW day^−1^); (**B**) Relative growth rate (ln DW increase day^−1^); (**C**) Carbon construction costs (mmol C g^−1^ DW) of 180-day-old *L. leucocephala* saplings grown in Veld Fertilizer Trial soils from Ukulinga Experimental Farm, South Africa. The values represent the mean ± SE, based on *n* = 5. Significant differences among treatments (*p* < 0.05) are denoted by different superscript letters.

**Figure 2 plants-10-02242-f002:**
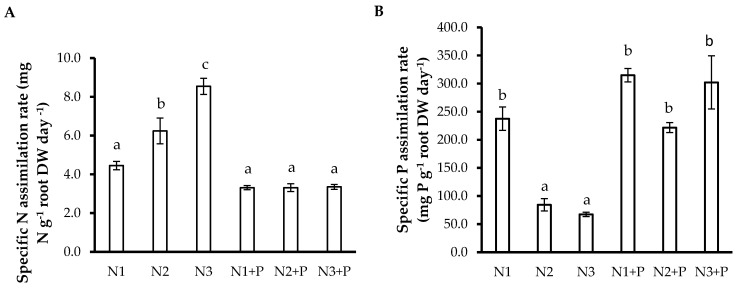
(**A**) Specific N assimilation rate (mg N g^−1^ root DW day^−1^); (**B**) Specific P assimilation rate (mg P g^−1^ root DW day^−1^**)**; (**C**) Specific N utilization rate (mg N g^−1^ plant DW day^−1^**)**; (**D**) Specific P utilization rate (mg P g^−1^ plant DW day^−1^) of 180-day-old *L. leucocephala* saplings grown in Veld Fertilizer Trial soils from Ukulinga Experimental Farm, South Africa. Values represent the means ± SE, based on *n* = 5. Significant differences among treatments (*p* < 0.05) are denoted by different superscript letters.

**Figure 3 plants-10-02242-f003:**
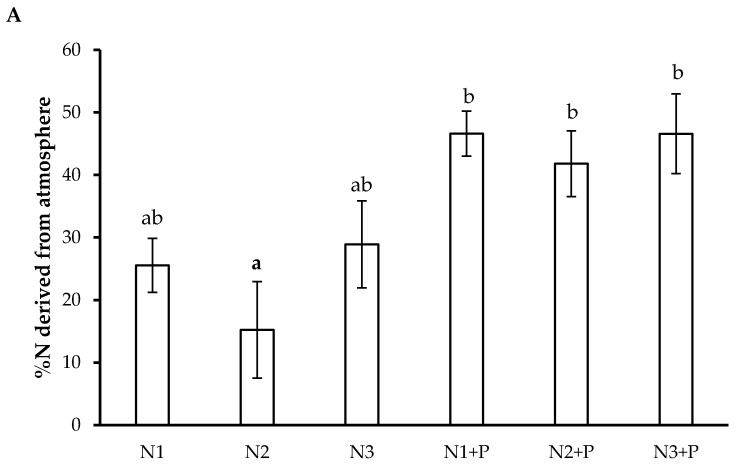
N preferences (**A**) Percentage N derived from the atmosphere, (**B**) Plant N concentration derived from atmosphere and Plant N concentration derived from soil (mmol N g^−1^ DW)) of 180-day-old *L. leucocephala* saplings grown in Veld Fertilizer Trial soils from Ukulinga Experimental Farm, South Africa. Values represent the means ± SE, based on *n* = 5. Significant differences among treatments (*p* < 0.05) are denoted by different superscript letters.

**Table 1 plants-10-02242-t001:** Biomass and mineral contents of 180-day-old *L. leucocephala* plants grown in Veld Fertilizer Trial soils from Ukulinga Experimental Farm, South Africa. Values represent the mean ± SE, based on *n* = 5. Significant differences among treatments are denoted by different superscript letters (*p* < 0.05).

Parameter	Treatment Trials					
	N1	N2	N3	N1 + P	N2 + P	N3 + P
Biomass (g DW)						
Total plant	1.285 ± 0.090 ^b^	1.154 ± 0.465 ^ab^	0.589 ± 0.042 ^a^	2.366 ± 0.236 ^c^	2.036 ± 0.236 ^bc^	2.891 ± 0.503 ^c^
Leaves	0.403 ± 0.034 ^b^	0.373 ± 0.126 ^a^	0.176 ± 0.008 ^a^	0.737 ± 0.106 ^b^	0.516 ± 0.087 ^b^	0.643 ± 0.098 ^b^
Shoot	0.305 ± 0.026 ^b^	0.272 ± 0.086 ^ab^	0.179 ± 0.026 ^a^	0.579 ± 0.056 ^c^	0.555 ± 0.075 ^c^	0.761 ± 0.099 ^c^
Roots	0.576 ± 0.040 ^bc^	0.509 ± 0.257 ^ab^	0.233 ± 0.020 ^a^	1.049 ± 0.122 ^d^	0.964 ± 0.098 ^cd^	1.487 ± 0.345 ^d^
Root–shoot ratio	0.819 ± 0.039 ^ab^	0.667 ± 0.100 ^a^	0.659 ± 0.039 ^a^	0.804 ± 0.063 ^ab^	0.924 ± 0.054 ^ab^	1.049 ± 0.138 ^b^
Mineral contents						
Total plant N (mmol N g^−1^ DW)	1.080 ± 0.051 ^a^	1.596 ± 0.168 ^b^	1.992 ± 0.096 ^c^	0.948 ± 0.027 ^a^	0.902 ± 0.055 ^a^	0.938 ± 0.034 ^a^
Standard corrected ^15^N/^14^N	2.550 ± 0.151 ^ab^	3.262 ± 0.177 ^b^	2.319 ± 0.230 ^ab^	1.099 ± 0.169 ^a^	1.430 ± 0.200 ^a^	1.101 ± 0.322 ^a^
Total plant P (µmol P g^−1^ DW)	60.92 ± 5.615 ^b^	21.37 ± 2.739 ^a^	15.63 ± 0.912 ^a^	76.60 ± 2.896 ^b^	59.22 ± 2.372 ^b^	82.99 ± 12.99 ^b^

## Data Availability

All raw data will be avilable upon request from the Dr Anathi Magadlela, the principal investigator of this work and can be contacted at anathimagadlela@icloud.com.

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
