# Peer review of "Nitrogen Source Preference and Growth Carbon Costs of Leucaena leucocephala (Lam.) de Wit Saplings in South African Grassland Soils"

_plants, 2021, doi:10.3390/plants10112242_

Round 1
Reviewer 1 Report
Line 63: Dot instead of comma.
Line 68: Here and in the rest of the text: “microbe symbiosis” should be “plant-microbe symbiosis”.
Lines 70-72: There is no novelty in this formulation of the hypothesis.
Lines 85-92: The meaning of these studies is not clear. Many questions arise. Was the same strain isolated from different locations? How is it proven that this is one strain? The similarity at the level below 95.5% does not allow attributing the strain to a specific species. Why did you choose and study these particular strains? With which strains from the NCBI were the isolated strains compared? There are hundreds of different types of bacteria in the soil, and many of them fix atmospheric nitrogen. This sample appears to be incomplete and unsubstantiated. This research has no scientific value.
Line 109: The use of the term “kinetics” to represent this data is questionable.
Lines 153-154: This speculation is not correct, since treatments N1 and N2 (low P supply) do not differ from treatments with high P supply.
Lines 159-160: It is not clear how this phrase can explain the results obtained. The differences in carbon construction costs are very small and there is no difference between treatments N1, N2, N3 and N2+P (Figure 1C)! Moreover, it seems that N1+P is similar to N2+P => please, check statistics!
Lines 164-177: In general terms, this reasoning is correct. However, the authors do not provide answers to the results obtained. Why did the plants form nodules in only one variant? Are nodule bacteria present in all soils? If NO, the plants did not choose nitrogen nutrition, but used whatever they could (soil and bacteria).
Lines 179-187: This reasoning also does not help to understand the results obtained.
Lines 188-191: There is no information in the article on how soil acidity (and some other parameters measured) affect the growth and nutrient uptake by pants.
Figure 3, N derived from atmosphere: Treatments N3 and N1+P both should be marked with “c”.
Lines 235-236: Indicate at what day the seedlings were emerged.
Lines 248-258: Why were the nodules sterilized 2 times? One time is enough. Why were the nodules stored at -4 ° C? At this temperature, water freezes and damages bacteria. This is the wrong method.
Lines 249; Why were the nodules crushed in 15% glycerol solution?
Lines 261-264: How many colonies were taken? How many phenotypes of colonies appeared at agar? What does “pure colonies” mean? Is it pure culture?
Lines 282-292: All the parameters used (R1, R2, t1, t2 and etc…) should be defined.
Line 306: What is difference between “the total organic nitrogen content” and “the total nitrogen content”?
Everywhere in the text: “sp.” should not be in italic.
Lines 336-341: The conclusion does not sound convincing. The sentence “plants maintained growth by relying on both atmospheric and soil derived N” is trivial, since all plant species do this. Why the isolated rhizobia are “highly effective”. On the contrary, they are not effective because almost did not affect the accumulation of atmospheric nitrogen by plants. “altering N sources” = This is an inappropriate conclusion, since there is no data on the dynamics of changes in the dominance of the nitrogen source in the same plants.
Author Response
Reviewer 1
Comments and Suggestions for Authors
Line 63: Dot instead of comma.
Author response: The comma has been replaced by a dot.
Line 68: Here and in the rest of the text: “microbe symbiosis” should be “plant-microbe symbiosis”.
Author response: This has been revised throughout the manuscript document
Lines 70-72: There is no novelty in this formulation of the hypothesis.
Author response: The novelty has been highlighted in the formulation of the hypothesis.
Lines 85-92: The meaning of these studies is not clear. Many questions arise. Was the same strain isolated from different locations? How is it proven that this is one strain? The similarity at the level below 95.5% does not allow attributing the strain to a specific species. Why did you choose and study these particular strains? With which strains from the NCBI were the isolated strains compared? There are hundreds of different types of bacteria in the soil, and many of them fix atmospheric nitrogen. This sample appears to be incomplete and unsubstantiated. This research has no scientific value.
Author response: Yes, the same strain was isolated from different locations. Also, the resulting sequences were edited and subjected BLASTN (National Center for Biotechnology Information, NCBI, https://www.ncbi.nlm.nih.gov/genbank/) to compare them to all of the other bacterial 16S rRNA sequences already in the database. So, we did not choose the strains but the results are based on the similarity of the current study resulting sequences and other bacterial 16S rRNA sequences already in the NCBI database.
Line 109: The use of the term “kinetics” to represent this data is questionable.
Author response: We have revised subtopic 2.3 from growth kinetics to growth rates and carbon construction costs
Lines 153-154: This speculation is not correct, since treatments N1 and N2 (low P supply) do not differ from treatments with high P supply.
Author response: We have revised this statement to reflect the statistical analysis.
Lines 159-160: It is not clear how this phrase can explain the results obtained. The differences in carbon construction costs are very small and there is no difference between treatments N1, N2, N3 and N2+P (Figure 1C)! Moreover, it seems that N1+P is similar to N2+P => please, check statistics!
Author response: We have revised this statement to reflect the statistical analysis.
Lines 164-177: In general terms, this reasoning is correct. However, the authors do not provide answers to the results obtained. Why did the plants form nodules in only one variant? Are nodule bacteria present in all soils? If NO, the plants did not choose nitrogen nutrition, but used whatever they could (soil and bacteria).
Author response: Only the L. leucocephala plants grown in N2+P soils were able to form a symbiotic association with the N fixing bacteria. The 16S rRNA gene revealed that the symbionts were various intermediate and fast- growing strains from the Mesorhizobium and Rhizobium sp. in plants grown in this treatment (Supplementary Table S1). Even though the Mesorhizobium and Rhizobium species were present in all soils. Non-nodulations in the L. leucocephala plants grown in the P deficient, low N and high N with P soils could be attributed to the low pH levels (pH (KCl) 4.12-4.67) observed in these soils compared to pH (KCl) 5.01 in N2+P soils. Extreme low pH levels have been reported to reduce approximately 90% of nodulation in various species such as, cowpea, pea, Lucerne and soybean [27] affecting both indeterminate and determinate nodules. For example, acidity dis-rupts the signaling mechanisms between the Rhizobium and the plant roots as explained in detail by [27-28].
Lines 179-187: This reasoning also does not help to understand the results obtained.
Author response: The 16S r RNA gene revealed that several species from the Mesorhizobium and Rhizobium genus inhabited the nodules of L. leucocephala grown in N2+P soils. Non-nodulations in the L. leucocephala plants grown in the P deficient, low N and high N with P soils could be attributed to the low pH levels (pH (KCl) 4.12-4.67) observed in these soils compared to pH (KCl) 5.01 in N2+P soils. Extreme low pH levels have been reported to reduce approximately 90% of nodulation in various species such as, cowpea, pea, Lucerne and soybean [27] affecting both indeterminate and determinate nodules. Acidity disrupts the signaling mechanisms between the Rhizobium and the plant roots as explained in detail by [27-28].
Lines 188-191: There is no information in the article on how soil acidity (and some other parameters measured) affect the growth and nutrient uptake by pants.
Author response: This information has been added in the manuscript discussion section.
Figure 3, N derived from atmosphere: Treatments N3 and N1+P both should be marked with “c”.
Author response: This has been changed in the manuscript as suggested by the reviewer.
Lines 235-236: Indicate at what day the seedlings were emerged.
Author response: Seedling emergence was observed after 10 days, then seedlings were transferred from petri-dishes to 15 cm diameter pots with experimental soils.
Lines 248-258: Why were the nodules sterilized 2 times? One time is enough. Why were the nodules stored at -4 ° C? At this temperature, water freezes and damages bacteria. This is the wrong method.
Author response: The second sterilization was to remove any contaminants that might have been introduced during the nodule transfer and storage. Though one time sterilization is recommended, this is a precautional practice we exercise in our lab to avoid any cross-contamination. Even though the equipment used during this process are thoroughly sterilized.
Lines 249; Why were the nodules crushed in 15% glycerol solution?
Author response: According to Somasegaran and Hoben (1994) Handbook for Rhizobia: Methods in Legume-Rhizobium Technology. Hielderberg, NY, Springer, they suggest that after sterilization, root nodules should be crushed in a microcentrifuge tube with 100ml of 15% glycerol. Ten ul of the turbid suspension with 15% glycerol solution should be streaked onto the surface of yeast mannitol agar medium. The turbid suspensions with 15% glycerol solution be stored in a freezer at -30 °C until isolations of rhizobia are complete. If the first isolation doesn’t succeed, the glycerol solutions are re-streaked onto fresh plates to obtain pure cultures.
Lines 261-264: How many colonies were taken? How many phenotypes of colonies appeared at agar? What does “pure colonies” mean? Is it pure culture?
Author response: The pure bacterial colonies/ cultures were randomly selected based on phenotypes (minimum 10 each culture with similar phenotypes) were amplified using a portion of 16-S rRNA gene, 27F (5’-AGAGTTTGATCCTGGCTCAG-3’) and 1492R (5’-GGTTACCTTGTTACGACTT-3’).
Lines 282-292: All the parameters used (R1, R2, t1, t2 and etc…) should be defined.
Author response: The parameters have been defined in the materials and methods section
Line 306: What is difference between “the total organic nitrogen content” and “the total nitrogen content”?
Author response: We analysed and considered total plant N concentrations and soil N concentration in the study and this data is presented in the results section
Everywhere in the text: “sp.” should not be in italic.
Author response: The text “sp.” has been changed throughout the manuscript
Lines 336-341: The conclusion does not sound convincing. The sentence “plants maintained growth by relying on both atmospheric and soil derived N” is trivial, since all plant species do this. Why the isolated rhizobia are “highly effective”. On the contrary, they are not effective because almost did not affect the accumulation of atmospheric nitrogen by plants. “altering N sources” = This is an inappropriate conclusion, since there is no data on the dynamics of changes in the dominance of the nitrogen source in the same plants.
Author response: The conclusion has been amended and aligned to the objectives and results of the study.
Author response: Thank you for the comments, they were really helpful in improving the manuscript.
Reviewer 2 Report
Review report of the manuscript entitled “Nitrogen source preference and growth carbon costs of Leucaena leucocephala (Lam.) de Wit in a South African grassland soil” by Sithole et al.
The paper fits well with the scope of the journal and encompasses some new aspects that would be of interest for the audience of the journal. The paper could be published after addressing the appended comments (major revision):
Major Comments:
- In Figure 1A and B: the growth rates show exceptionally high SD values but no justification for the same has been provided in the discussion.
- In Figure 1C: There is insignificant change in carbon construction costs in all the treatments. Please provide correlate the same the phenomenally good growth rates that were observed during test period.
- On comparative analysis of Figure 3A and 3C difference in Net N utilization is irrational, specifically for treatments N1+P, N2+P and N3+P. Please discuss the rationale behind the net negative difference.
- The biggest limitation of this MS is its weak discussion. Please make sure to discuss all the results in light of recent studies.
- Limitations of the study should be included in the conclusion section.
Minor Comments:
- The tile of the MS grammatical errors, please revise the same
- Section 4: Materials and Methods should placed after the introduction section
- Line 205: The closing bracket is missing
- In section 1: Please denote what type of soil was studied.
- Line 284 and further, all equations must be numbered, like Eq.1, Eq.2 etc. It should be like:
SNAR = (N2 – N1 / t2 – t1) * [(loge R2 - loge R1) / (R2 – R1)] Eq.1
- In section 4.7.2: Relative growth rate should not be separated, combine it with the previous one (4.7.1)
- Statistical analysis, please specify the level of confidence with which the statistical processing was carried out
- Line 211: “fertilizer were applied two times a year” and Line 213 “applied once a year”. This is major technical error and should be rectified responsibly. Also, please indicate when exactly the fertilizers were applied i.e. in which season and which stage of plant growth.
- Line 111: “…in other soils”. If I understand correctly, the soil was the same, the content of N and P was different, is it right? Please revise throughout the MS (Line 114 etc.)
Author Response
Reviewer 2
The paper fits well with the scope of the journal and encompasses some new aspects that would be of interest for the audience of the journal. The paper could be published after addressing the appended comments (major revision):
Major Comments:
In Figure 1A and B: the growth rates show exceptionally high SD values but no justification for the same has been provided in the discussion.
Author response: Justification and highlights of this have been discussed in the discussion section of the manuscript.
In Figure 1C: There is insignificant change in carbon construction costs in all the treatments. Please provide correlate the same the phenomenally good growth rates that were observed during test period.
Author response: This has been corrected in the results section as suggested by reviewer.
On comparative analysis of Figure 3A and 3C difference in Net N utilization is irrational, specifically for treatments N1+P, N2+P and N3+P. Please discuss the rationale behind the net negative difference.
Author response: This has been thoroughly described and discussed in the discussion section as suggested by the reviewer.
The biggest limitation of this MS is its weak discussion. Please make sure to discuss all the results in light of recent studies. Limitations of the study should be included in the conclusion section.
Author response: The discussion has been rewritten and the conclusion has been amended and aligned to the objectives and results of the study.
Minor Comments:
The tile of the MS grammatical errors, please revise the same
Author response: The grammatical errors have been corrected throughout the manuscript as suggested by the reviewer.
Section 4: Materials and Methods should placed after the introduction section
Author response: According to Plants manuscript template available in the Plants website which was downloaded and used for this manuscript, materials and methods should follow the discussion.
Line 205: The closing bracket is missing
Author response: The closing bracket has been added.
In section 1: Please denote what type of soil was studied.
Author response: Top soil samples (0-30 cm depth and 2 m apart)
Line 284 and further, all equations must be numbered, like Eq.1, Eq.2 etc. It should be like:
SNAR = (N2 – N1 / t2 – t1) * [(loge R2 - loge R1) / (R2 – R1)] Eq.1
Author response: All equations in the materials and methods section of the manuscript have been numbered.
In section 4.7.2: Relative growth rate should not be separated, combine it with the previous one (4.7.1)
Author response: Relative growth rate was combine with 4.7.1 calculations as suggested by reviewer
Statistical analysis, please specify the level of confidence with which the statistical processing was carried out
Author response: The level of confidence has been included in the statistical analysis description section, in all the figure legends and table description topics.
Line 211: “fertilizer were applied two times a year” and Line 213 “applied once a year”. This is major technical error and should be rectified responsibly. Also, please indicate when exactly the fertilizers were applied i.e. in which season and which stage of plant growth.
Author response: The Ukulinga Grassland Nutrient Experiment (UGNE) is located at Ukulinga research farm of the University of KwaZulu, Pietermaritzburg, South Africa (29° 24 E, 30° 24 S). was initiated in 1951 through the manipulation of nitrogen (N), phosphorus (P). The objectives of the long-term VFT were to increase productivity of fodder, which was of interest, to determine the effect of nutrient addition (N and P) on the production, crude protein content and composition of a grassland. There were initially 96 plots from years 1951-2019 and each plot was 9.0 x 2.7 metres (m) in size with a 1 m spacing between plots. The experiment was replicated in three blocks, each block containing 32 plots, resulting in a 4 x 23 factorial design. Three levels of 28% LAN (N1 = 210 kg/ha/season; N2 = 421 kg/ha/season and N3 = 632 kg/ha/season) fertilizer. In addition, the three N levels were also applied in combination with one level of 11.3% superphosphate (336 kg/ha/season) (N1+P, N2+P and N3+P).
References
Tsvuura, Z. & Kirkman, K. P. Yield and species composition of a mesic grassland savanna in South Africa are influenced by long-term nutrient addition. Austral Ecol. 38, 959–970 (2013).
Morris, C. D. & Fynn, R. W. S. The Ukulinga long-term grassland trials: reaping the fruits of meticulous, patient research. Bull. Grassl. Soc. South Africa 11, 7–22 (2001).
Author response: Thank you for the comments, they were really helpful in improving the manuscript.
Reviewer 3 Report
The authors in their article “Nitrogen source preference and growth carbon costs of Leucaena leucocephala (Lam.) de Wit in a South African grassland soil” investigated the microbe symbiosis, plant nutrition, C costs, and biomass accumulation in L. leucocephala grown in acidic soils with varying N and P nutrient status. The experimental design is well-organized, and the presented dataset is interesting. Nevertheless, there are some parts of this article that should be further improved before this article can be considered suitable for publication.
- One of the most interesting and principal questions of the initial hypothesis was that leucocephala will establish symbiosis with multiple and more efficient N-fixing bacteria. The authors have applied the proper methods to try to identify these bacteria, but we find the results to be summarized in a supplementary table, and only a small part of the work to be about this specific topic. In my opinion, this is one of the topics that the authors should choose to analyze and present in-depth. Especially since the conclusions are referring to many times in the “symbiosis”. Further analyses are needed to present in a clear way certain differences.
- The introduction is very short. I am not in favor of long introductions, but the current form of the introduction is not fully supporting the need for the specific study
- Lines 206-209 describing the replication of the experiment are not written in a clear way. Please rephrase
- In the M&M section, it is not clear when the samples were collected. The authors in line 242 mentioned that “The initial and final harvests took place 30 and 180 days after seedling emergence” but no graphs are presenting 30-day results. The authors must clarify when the samplings took place in the initial experimental design
- In graph 3b capital letters should be used in either one of the two cases to show the differences (either above the white or the gray bars).
- Lines 330-332. Since the authors applied more parametric and non-parametric methods to analyze their results, it is important to mention in each table or figure capture the exact type of analysis that they applied.
Author Response
Reviewer 3
Comments and Suggestions for Authors
The authors in their article “Nitrogen source preference and growth carbon costs of Leucaena leucocephala (Lam.) de Wit in a South African grassland soil” investigated the microbe symbiosis, plant nutrition, C costs, and biomass accumulation in L. leucocephala grown in acidic soils with varying N and P nutrient status. The experimental design is well-organized, and the presented dataset is interesting. Nevertheless, there are some parts of this article that should be further improved before this article can be considered suitable for publication.
One of the most interesting and principal questions of the initial hypothesis was that leucocephala will establish symbiosis with multiple and more efficient N-fixing bacteria. The authors have applied the proper methods to try to identify these bacteria, but we find the results to be summarized in a supplementary table, and only a small part of the work to be about this specific topic. In my opinion, this is one of the topics that the authors should choose to analyze and present in-depth. Especially since the conclusions are referring to many times in the “symbiosis”. Further analyses are needed to present in a clear way certain differences.
Author response: A more detailed study solely focusing on the experimental soil nutrition, soil microbial diversity, phosphorus, nitrogen and carbon soil enzyme activity and microbial nutrient solubilizing efficiency has been submitted and under review in Soil Biology and Biochemistry. Therefore, as the authors we decided to include the soil nutrition and soil microbial data, which forms the baseline data of this study as a supplementary figure to avoid self-plagiarism which against scientific ethics.
The introduction is very short. I am not in favor of long introductions, but the current form of the introduction is not fully supporting the need for the specific study
Author response: More literature was included in the introduction section as suggested by the reviewer.
Lines 206-209 describing the replication of the experiment are not written in a clear way. Please rephrase
Author response: The section describing the replication of the experiment has been rewritten for clarity.
In the M&M section, it is not clear when the samples were collected.
Author response: The soil samples were collected in February 2019, which is mid-summer in South Africa.
The authors in line 242 mentioned that “The initial and final harvests took place 30 and 180 days after seedling emergence” but no graphs are presenting 30-day results. The authors must clarify when the samplings took place in the initial experimental design
Author response: The initial harvest of five saplings from each treatment for the initial values required in the growth calculations took place 30 days. Because we wanted to calculate growth overtime, we needed initial plant samples and final plant samples to enable us to do the calculations.
In graph 3b capital letters should be used in either one of the two cases to show the differences (either above the white or the gray bars).
Author response: In graph 3B, the letters have been capitalized in the grey bars, N derived from atmosphere.
Lines 330-332. Since the authors applied more parametric and non-parametric methods to analyze their results, it is important to mention in each table or figure capture the exact type of analysis that they applied.
Author response: The different types of analysis have been specified in detail in the tables and figures as suggested by reviewer.
Author response: Thank you for the comments, they were really helpful in improving the manuscript.
Reviewer 4 Report
The publication brings interesting information about the nitrogen and phosphorus uptake capacity and seedling development of Leucaena leucocephala, an invasive plant
The paper requires additions:
- in the subject of the paper it should be indicated that seedlings of L. leucocephala were studied;
- In the research methodology, the description of the research methods should be supplemented according to the remarks in the manuscript;
- in the discussion of the results, the abundance of humus, P,N and K should be characterized - whether the amounts of these elements are high or low in comparison with the soil abundance standards. Moreover, the granulometric composition of soil is important for the assessment of soil properties - to be completed;
- the discussion of the results is based only on 7 publications, which are not directly related to L. leucocephala - the discussion should be supplemented according to remarks included in the manuscript;
- a paragraph on the adaptive and invasive capabilities of L. leucocephala should be included in the conclusions;
- Other minor comments are included in the text of the manuscript

Author Response
Reviewer 4
Comments and Suggestions for Authors
The publication brings interesting information about the nitrogen and phosphorus uptake capacity and seedling development of Leucaena leucocephala, an invasive plant
The paper requires additions:
in the subject of the paper it should be indicated that seedlings of L. leucocephala were studied;
Author response: This has been specified in the title of the manuscript
In the research methodology, the description of the research methods should be supplemented according to the remarks in the manuscript;
Author response: The manuscript sections have been amended as suggested by the reviewer and the discussion has been rewritten according to the remarks in the manuscript.
in the discussion of the results, the abundance of humus, P, N and K should be characterized - whether the amounts of these elements are high or low in comparison with the soil abundance standards.
Author response: The discussion has been rewritten to include the soil nutrition as this forms the baseline data of the study, however, this is done in such a way that this baseline data is integrated with plant-microbe symbiosis, plant performance and plant nutrition. This is because a more detailed study solely focusing on the experimental soil nutrition, soil microbial diversity, phosphorus, nitrogen and carbon soil enzyme activity and microbial nutrient solubilizing efficiency has been submitted and under review in Soil Biology and Biochemistry. Therefore, as the authors we decided to avoid self-plagiarism which against scientific ethics, we carefully integrated this data carefully with these major factors to avoid this.
Moreover, the granulometric composition of soil is important for the assessment of soil properties - to be completed;
Author response: Thank you for suggesting the granulometric composition analysis, we have tried to look for an external lab to do this for us in the country as we don’t have the capacity to do this analysis in our lab with no luck to finding one, however, we will consider this analysis for follow-up studies.
the discussion of the results is based only on 7 publications, which are not directly related to L. leucocephala - the discussion should be supplemented according to remarks included in the manuscript;
Author response: The discussion has been rewritten and highlights of the results were thoroughly discussed to support our conclusions. Also, though no studies have specifically looked at the aspects of soil nutrition and plant microbe symbiosis in L. leucocephala and plant nutrition. We have tried to integrate our discussion with more relevant and updated literature on other legume plant studies looking at these different aspects.
a paragraph on the adaptive and invasive capabilities of L. leucocephala should be included in the conclusions;
Author response: A section of the conclusion highlighting the adaptive and invasive capabilities of L. leucocephala as presented in the results of this study has been added.
Other minor comments are included in the text of the manuscript
Author response: The suggested minor comments have been incorporated in the manuscript.
Author response: Thank you for the comments, they were really helpful in improving the manuscript.
Round 2
Reviewer 2 Report
NIL
Author Response
Thank you for the comments, they were really helpful in improving the manuscript.Reviewer 3 Report
The authors have addressed all my comments. Even though I believe that this article would be much better if more information regarding the soil microbial diversity was added, I still find it suitable for publication
Author Response
Thank you for the comments, they were really helpful in improving the manuscript.